# A Novel Method for Creating a Synthetic L-DOPA Proteome and In Vitro Evidence of Incorporation

**DOI:** 10.3390/proteomes9020024

**Published:** 2021-05-24

**Authors:** Joel Ricky Steele, Natalie Strange, Kenneth J. Rodgers, Matthew P. Padula

**Affiliations:** 1Proteomics Core Facility and School of Life Sciences, The University of Technology Sydney, Ultimo, NSW 2007, Australia; matthew.padula@uts.edu.au; 2Neurotoxin Research Group, School of Life Sciences, The University of Technology Sydney, Ultimo, NSW 2007, Australia; kenneth.rodgers@uts.edu.au; 3School of Life Sciences, The University of Technology Sydney, Ultimo, NSW 2007, Australia; natalie.strange@student.uts.edu.au

**Keywords:** misincorporation, post-translational modifications, PTM, Levodopa, L-DOPA

## Abstract

Proteinopathies are protein misfolding diseases that have an underlying factor that affects the conformation of proteoforms. A factor hypothesised to play a role in these diseases is the incorporation of non-protein amino acids into proteins, with a key example being the therapeutic drug levodopa. The presence of levodopa as a protein constituent has been explored in several studies, but it has not been examined in a global proteomic manner. This paper provides a proof-of-concept method for enzymatically creating levodopa-containing proteins using the enzyme tyrosinase and provides spectral evidence of in vitro incorporation in addition to the induction of the unfolded protein response due to levodopa.

## 1. Introduction

The non-protein amino acid (NPAA) L-3,4-dihydroxyphenylalanine (L-DOPA), commercially known as ‘levodopa’, is used to restore dopamine levels in the damaged substantia nigra of Parkinson’s disease (PD) patients [1] although it has also been hypothesised to accelerate PD pathology and neurodegeneration [2] by causing neuronal toxicity [2,3,4,5,6,7,8,9,10,11,12,13,14,15,16]. The mechanisms of L-DOPA-induced neurodegeneration include induction of oxidative stress and the misincorporation of L-DOPA into proteins in place of L-tyrosine, resulting in protein misfolding and the formation of proteolysis-resistant aggregates [3,5,7,17,18,19]. To date, in vitro studies have not examined or identified the proteins containing incorporated L-DOPA and their potential role in disease pathology, with Parkinson’s disease datasets being a prime candidate [20].

Protein-bound DOPA (PB-DOPA) can form via several mechanisms, including direct incorporation of free L-DOPA in place of tyrosine into a growing polypeptide chain [21]. Alternatively, L-DOPA can be formed via hydroxyl attack on tyrosine residues by a reactive oxygen species (ROS) [21,22,23,24,25], independent of free L-DOPA. Similarly, hydroxyl radical attack on the phenyl ring of phenylalanine residues forms isomers of tyrosine [22,26], the position of the hydroxyl group of these isomers (meta and ortho) differing from that of the protein amino acid L-tyrosine (para-tyrosine) (Figure 1). It is known that free L-DOPA can induce oxidative stress, resulting in ROS generation via interaction with iron and copper to produce Fenton reaction products [27], which leads to further hydroxylation of free or protein incorporated tyrosine, thereby producing free and PB-DOPA [5,22].

The misincorporation rate of canonical amino acids in general is reported to be approximately one in 10,000; this low level of misincorporation makes it difficult to detect DOPA incorporation or other misincorporated amino acids [18,21,28,29]. However, it is possible that misincorporation levels of NPAAs could be higher since they can escape proofreading mechanisms that have evolved to detect the mistaken incorporation of a canonical amino acid. As proteomics utilises mass spectrometers, which are concentration-dependent detectors, methodologies should be applied to enhance the ability to detect these low-abundance species (for a comprehensive review, see Steele et al., 2021 [28]). The most sensitive way to detect low-abundance species in exploratory proteomics is the use of data-independent acquisition (DIA), but this requires prior identification of the peptide species and its fragments for a searchable spectral library, especially in instances of modified peptides [29]. By producing synthetic peptides for subsequent proteomic analysis and library inclusion, it becomes possible to identify very low levels of the modified proteoform. However, the production of such a synthetic library is prohibitively expensive and the libraries currently available do not contain NPAA modifications [30].

The best measure of the global oxidative state of a proteome is the level of methionine oxidation, which occurs via the addition of a single oxygen atom, creating methionine sulfoxide with a mass shift of +15.99 Da, homologous to that of hydroxylation [31,32]. However, protein extraction and preparation can increase the likelihood of forming oxidised methionine and even electrospray ionisation can result in this modification [29,33,34]. Due to the multiple possible end-points of hydroxyl attack, the identification of oxidised phenylalanine may help in determining if PB-DOPA has been formed from incorporation or from hydroxyl attack. Within plant and animal tissues, the conversion of tyrosine to L-DOPA can be carried out by the enzyme tyrosinase, which has also been used in industrial synthesis of L-DOPA and is essential to the production of dopamine in the human brain [33,34]. Unlike oxidation reactions, the enzyme tyrosinase has the capacity to generate peptide-bonded DOPA (from PB-tyrosine conversion) without generating other oxidised amino acid residues such as ortho- and meta-tyrosine.

To overcome the poor detection of low-ion-abundance peptides from PB-DOPA proteoforms, a method for creating a positive control spectral library needs to be produced to enable future application to clinical samples by enabling DIA analyses. A method for generating a positive control spectral library and avoiding the high cost of chemically synthesising a DOPA-containing peptide library is to use the enzyme tyrosinase to specifically modify tyrosine residues to L-DOPA [34,35]. The aim of this work is to establish the use of tyrosinase to convert proteomes, enabling subsequent analyses. Furthermore, we explore the effects of DOPA on a neuronal cell line and investigate the proteins that contain PB-DOPA following treatment with L-DOPA.

## 2. Materials and Methods

### 2.1. Materials

The following reagents were sourced from Sigma-Aldrich (Saint Louis, MI, USA): L-DOPA (Cat #D9628); urea (Cat #U5378); thiourea (Cat #T7875); C7BzO (Cat #C0856); Dulbecco’s Modified Eagle Medium (DMEM; Cat#D6429) (New York, NY, USA); cOmplete™ Protease Inhibitor Cocktail (Cat #11697498001) (Roche, USA); tris(2-carboxyethyl)phosphine (TCEP; Cat #C4706); ammonium bicarbonate (Cat #A6141); Empore™ Extraction Disk Cartridge C18 (Cat #66873-U) (Supelco, Bellefonte, PA, USA); acetonitrile hypergrade for LC–MS LiChrosolv^®^ (Cat #1.00029) (Truganina, Australia); triflouroacetic acid (Cat #T6508); tyrosinase (Cat #T3824); boric acid (Cat #1.00165); sodium hyrdoxide (Cat # 221465); ascorbic acid (Cat # A4544); hydroxylamine solution (Cat # 467804). The following reagents were sourced from Thermo Fisher: TrypLE™ Express Enzyme (Cat #12604013) (Gibco, New York, NY, USA); Pierce™ BCA Protein Assay Kit (Cat #23225) (Thermo Scientific, Rockford, IL, USA). The UCP-5 peptide was synthesised by the Payne Laboratory with sequence: EEGVLALYSGIAPALLR.

### 2.2. Tyrosinase Method Optimisation

The neuroblastoma cell line SH-SY5Y (ATCC #CRL-2266) was utilised to optimise a microscale proteome tyrosinase conversion. Six T175 flasks were treated in Dulbecco’s Modified Eagle Medium with 10% *v/v* fetal bovine serum (FBS) at 37 °C with 5% CO_2_. Three flasks were treated with 500 µM L-DOPA-containing media for 24 h; the DOPA addition was from a 3 mM stock in 10 mM HCl with control flasks receiving an equal amount of sterilised HCl solution. Cells were washed three times with PBS, de-adhered using TrypLE (Gibco, New York, NY, USA), pelleted at 1500 rcf and snap frozen in liquid nitrogen.

### 2.3. Protein Extraction and Processing

Cell pellets were solubilised in UTC7 (8 M urea, 2 M thiourea, 0.1% *w*/*v* C7BzO) with cOmplete protease inhibitors (Roche). Pellets were kept on ice when adding solubilisation buffer, then subjected to probe sonication with 80–100% intensity for 3 × 30 s rounds with resting on ice between rounds. Samples were centrifuged at 10,000× *g* to pellet insoluble material and the supernatant transferred to fresh tubes. Protein yields were then assayed via gel-based densitometry [36] and BCA assay (Thermo Fisher, Rockford, MA, USA). Protein samples were then reduced using 5 mM tris(2-carboxyethyl)phosphine (TCEP) to break disulfide linkages and linearise proteins; the cysteines were then blocked with 20 mM acrylamide monomers for 90 min. Protein digestion was performed on 100 µg of sample with the UTC7 solution diluted by 8-fold with 100 mM ammonium bicarbonate solution to enable the activity of trypsin by ensuring the pH is above 8.0 and the concentration of urea is below 1 M. Peptides were cleaned of contaminating buffer components using C18 solid-phase extraction columns (Empore™ Extraction Disk Cartridge). Briefly, the columns were activated with 100% acetonitrile (ACN) then equilibrated with 2% *v*/*v* ACN with 0.2% *v*/*v* trifluoracetic acid (TFA). Samples were then applied to the equilibrated column and then washed with 2% *v*/*v* ACN; elution was achieved with 50% *v*/*v* ACN, 0.1% *v*/*v* TFA. The organic solvent was then removed by rotary vacuum-assisted evaporation under non-heating conditions. These samples were then analysed utilising a LC–MS/MS on a QExactive Plus (Thermo Fisher Scientific, Bremen, Germany) [37].

### 2.4. Control Peptide Lysate Preparation

For the creation of a L-DOPA-positive control proteome, the three non-treated replicate digests were pooled for processing, allowing comparison of the controls against the L-DOPA-treated sample in terms of percentage conversion. All samples underwent the same methodology until the point before injection onto the mass spectrometer. The peptide solutions were aliquoted and frozen at −80 °C, then lyophilised in a freeze drier (Christ). These peptides were then stored at −20 °C until required.

#### 2.4.1. Tyrosinase Conversion Proof of Concept

Previous work investigating tyrosinase conversion used much larger amounts of material than would normally be available for proteomic studies. Thus, a proof-of-concept microscale experiment was performed utilising a synthetic peptide sequence from UCP-5, a mitochondrial protein implicated in Parkinson’s disease [38]. Following the method by Ito et al. 1984 [35], conversion efficiencies were so low that detectable conversion on an entire proteome was not possible.

#### 2.4.2. Microscale Whole Proteome Tyrosinase Conversion

An optimised method for tyrosinase conversion adapted from Ceinska et al. 2016 [34] was applied to the control peptide lysate. Molar concentrations for the reaction were based on an average peptide weight of 1 kDa (https://www.bioline.com/media/calculator/01_04.html) (accessed on 1 April 2018). The reaction buffer final constitution was of 0.5 mM borate buffer adjusted to pH 7.0 with NaOH, 2 mM ascorbic acid, 6.7 mM hydroxylamine and varying ratios of tyrosinase to peptide digest. For concentration-dependent scaling down, 1 mM (100 µg) of peptide was reacted with differing ratios of tyrosinase (10:1, 2:1, 1:1) in 100 µL reaction volumes. The most detrimental factor reported for the industrialised conversion method is the lack of aeration in the reaction vessel and the exhaustion of free oxygen during back conversions of deoxy-tyrosine and creation of the hydroxylated residues. To prevent this, the micro-scaling of the reaction inside of a microtube combined with low-speed desktop vortexing ensures aeration of the reaction. The reaction was performed at 30 °C for 240 min. To remove the tyrosinase from the peptide proteome to be analysed, a 3 kDa molecular weight cut off spin filter (Pall) was used as the intact 112 kDa tyrosinase enzyme is retained and the peptides freely pass through. This sample was then diluted to the correct concentration for injection onto the QExactive Plus mass spectrometry platform with interfering buffer components washed away during peptide trapping before analysis.

### 2.5. MALDI Analysis of Synthetic UCP-5 Conversion

Native and converted UCP-5 peptides were spotted at a volume of 1 µL combined with 1 µL of 5 mg/mL CHCA dissolved in 50% *v*/*v* acetonitrile, 0.1% *v*/*v* TFA, 10 mM NH_4_H_2_PO_4_ onto a clean 386-well OptiTOF target plate (AB Sciex) and allowed to dry. Peptide samples were then analysed using a 5800 MALDI-TOF/TOF (AB Sciex) mass spectrometer in positive ion reflector mode. Laser intensity was set to 2600 for MS parent ion scans and 3000 for MSMS fragmentation ion scans; 400 laser shots were averaged for MS scans and up to 1250 shots were averaged for MSMS scan with the Dynamic Exit algorithm selected. MS parent ion scans were calibrated using the TOF/TOF standards while MS/MS fragmentation ion scans were calibrated using the fragments of Glu-Fibrinopeptide B present in the TOF/TOF standards mixture. The resulting MS and MS/MS spectral data were then converted to Mascot Generic Format using T2D Extractor and the data searched with Mascot (provided by the Australian Proteomics Computational Facility, hosted by the Walter and Eliza Hall Institute for Medical Research Systems Biology Mascot Server) against the Human Proteome with variable modification Y(+15.99). Theoretical fragments were generated utilising the fragment ion calculator hosted by http://db.systemsbiology.net (accessed on 20 August 2017).

### 2.6. Q-Exactive Plus LC–MS/MS

Peptides were analysed using a QExactive Plus mass spectrometer (Thermo Fisher Scientific) and a M-Class Acquity liquid chromatograph (Waters). Peptides were subject to electrospray ionisation (ESI) at 2.6 kV using a PicoFrit column (75 mm ID × 300 mm; New Objective, Woburn, MA, USA) packed with Daisogel SP-ODS-BIO C18 resin (3 µm) (Osaka Soda Co, Osaka, Japan). Data-dependent acquisition was performed using the top 12 highest-abundance ions with dynamic exclusion of precursors for 30 s post-fragmentation, resolution was set to 70,000 for a survey scan range of 350*_m_*_/*z*_–1500*_m_*_/*z*_, with an AGC target of 1 × 10^6^ and a maximum injection time (I.T) of 100 ms, fragmentation scans were acquired at a resolution of 17,500, with an I.T of 50 ms, an isolation window of 1.4*_m_*_/*z*_, a mass range beginning at 50_m/z_ and a normalized collision energy (NCE) of 28. Prior to ESI, 1 µg of peptide samples was loaded on to a nanoEase Symmetry C18 trapping column (180 µm × 20 mm), analytical chromatographic separation was performed utilising buffer A containing 99.8% ultra-pure water with 0.2% formic acid and buffer B containing 99.8% acetonitrile and 0.2% formic acid. Separation occurred over a 140 min gradient at a constant flow rate of 300 nL/min, and gradient steps were linear with buffer A (95, 95, 60, 20, 99, 99) at minute (0, 1, 121, 123, 125, 127, 150).

### 2.7. Data Analysis

Raw files were loaded into PEAKS Studio (v10.6, build 2020 1015) [39]. Searches were performed using the following parameters: a parent mass tolerance of 20 ppm; a fragment tolerance of 0.02 Da; semi-specific trypsin with three missed cleavages; de novo sequencing with a maximum of modifications three per peptide—fixed: propionamide (C), variable: carbamylation (K and N-term, identified on a first pass PTM search) and oxidation (M, Y, F). PEAKS database searching was performed against the human proteome containing isoforms indexed on the 3 September 2017 (71,540 entries combined from TREMBL and SwissProt). Proteins containing greater than 10,000 amino acids were removed, and standard contaminants were also included. PEAKS PTM searching was performed on the previous database search results using the inbuilt 312 modifications. Results were filtered to include de novo scores ≥50, peptides -10lgP score of ≥16.1, and a 1% FDR, reporting 40,752 identified PSMs. Proteins at 1% FDR with one unique peptide resulted in a total of 9462 identified proteins. PTMs with an Ascore above 1 (/1000) were retained.

The control and treatment technical injections were also searched using MaxQuant (v1.6.14.0) against the same database and common contaminants. All parameters can be found within the MaxQuant summary text file (PRIDE: PXD025759)**.** Briefly, parameters were set to a minimum peptide length of 7, fixed modification propionamide (C), variable modifications of oxidation (M, Y, F) and N-terminal acetylation. MaxQuant LFQ was performed with a minimum of two unique peptides used for protein identification and one unique plus razor peptide used for quantitation with a minimum ratio of two. An FDR of 0.01 was used for PSM protein and site, with match between runs used with a matching window of 0.7 min and an alignment window of 20 min.

MaxQuant output files were loaded into the Mass Dynamics webserver ([40], https://app.massdynamics.com/) (accessed on 1 March 2021) for analysis, a public link to this project has been generated https://app.massdynamics.com/p/dfbb55e9-0a6d-450e-8e8e-3f0c070e5bf9 (accessed on 1 March 2021). A total of 2047 protein groups were identified, and 1846 protein groups quantified. Proteins with an adjusted FDR of less than 5% and 2-fold up or down in abundance were used for further pathway analysis (Appendix A). Pathway analysis was performed using the pathways tool that queries Reactome repository [41]. Proteins significantly changing in abundance (adjusted *p*-value of <0.05 and a log_2_-fold change >1) were exported from Mass Dynamics project and annotated using UniProt’s ID mapping function. Further analysis of protein interactions between up and down proteins were also performed within STRING (v11.0) [42]; medium confidence was used with all interaction data sources enabled.

## 3. Results

### 3.1. Tyrosinase Conversion of Synthetic Peptide UCP5

To demonstrate that tyrosinase could be used to convert tyrosine residues in a peptide to DOPA, the synthetic peptide UCP5 (SLC25A14) (mitochondrial uncoupling protein) was chosen for conversion due to its implication in the maintenance of mitochondrial stability and its exploration in models of oxidative stress in SH-SY5Y cells [20]. The tryptic peptide sequence used—EEGVLALYSGIAPALLR—was successfully converted to contain DOPA by the method of Ito et al. [35]. Figure 2 shows the theoretical fragments generated by the native peptide and the L-DOPA-containing peptide with spectral evidence confirming this pattern shown underneath.

### 3.2. Proteome Analysis

The method of Ito et al. [35] was not successful when applied on a proteome level (data not shown). Thus, the tyrosinase conversion method outlined by Cieńska et al. [34] was scaled down and applied to the SH-SY5Y control proteome. To assess the off-target effects, the percentages of tyrosine (Y), methionine (M) and phenylalanine (F) residues displaying oxidation are presented in Figure 3. Regardless of residue or peptide sequence, the percentage oxidation displays a similar pattern for each sample. The DOPA-treated biological samples display a significant increase in methionine oxidation (~26%) versus the control (~1%), with a non-statistically significant increase in the amount of DOPA and oxidised phenylalanine. All three tyrosinase amounts used for treatment resulted in a >10% increase in DOPA-containing peptides, with the 1:1 conversion having a higher specific L-DOPA formation noted by the decreased amount of phenylalanine oxidation. The conversion process led to >75% methionine oxidation across all three enzyme:substrate ratios and, regardless of which ratio was used, DOPA formation is significantly lower than the 95% conversion achieved on free tyrosine reported in the unscaled method [34]. However, a significantly lower number of spectra were detected following treatment, and even less following the conversion process, despite an equal concentration of protein being processed for each sample. A label-free quantitative analysis was performed in PEAKS Studio to generate the total ion count normalization factors displayed in Figure 4, which indicates a notable decrease in the amount of sample ionised in treatments and conversions relative to control.

### 3.3. Proteoform Analysis

Proteoforms containing DOPA were generated by incubation of cells with L-DOPA in cell culture or by conversion of tyrosine present in tryptic peptides generated from control cell lysates to DOPA using the enzyme tyrosinase. From the output data, DOPA-containing peptide sequences were identified in untreated control, DOPA-treated or tyrosinase-converted samples (Table 1). The identification was based on feature detection and ‘match between runs’ or peptide identification transference rather than peptide identification in each injection, and sequences that could not be confidently attributed to a condition were discarded. From the control samples, only 34 peptide sequences could be accurately identified as containing DOPA whilst for the DOPA-treated samples there is spectral evidence for 101 sequences containing DOPA. The tyrosinase conversion produced 532 DOPA-containing peptide sequences. The sequences and associated protein identifications can be found in Appendix A. For the DOPA-containing peptide sequences, protein identifications were extracted and compiled, and 75 active UniProt entries were found in the DOPA-treated cells and 37 in the control. Overlap of proteins containing DOPA between control and DOPA treatment was performed (Appendix A), resulting in 30 proteins being unique to control and 68 unique to treatment. Four proteins were identified in both groups: valine--tRNA ligase; 26S proteasome regulatory subunit 6A; calreticulin; tubulin beta-4A chain; clathrin heavy chain 1; clathrin heavy chain; and heat shock protein HSP 90-beta. Analysis of the biological process GO terms unique to control identified four key terms that covered the identifications; response to stress; organelle organization; post-transcriptional regulation of gene expression; and sulfur compound biosynthetic process. Analysis of the biological process GO terms of the unique to DOPA with strongest FDR identified; protein folding; establishment of localisation; transport; and regulation of biological quality.

When analysing the tyrosinase-converted samples to identify DOPA-containing peptides, four peptides were identified in the biological samples from quality fragment data generated in the converted samples (Table 2), which would have otherwise not been detected with confidence.

### 3.4. Biological Insights from Pathway Analysis

Quantitative analysis was performed on the control vs. treated samples using MaxQuant. Pathway analysis was subsequently performed using the Mass Dynamics webserver, with the volcano plot shown in Figure 5. All proteins with a log_2_-fold change ≥1 and adjusted *p*-value < 0.05 were used for subsequent pathway analyses. The project results can be accessed and interacted with by the public link provided. The statistically significantly enriched pathways (<5% FDR) can be found in the mass dynamic project, which shows proteins involved in the metabolism of RNA and its processing to be decreased in abundance.

Quantitative analysis of the proteome quantified 1846 protein groups with 390 significantly changing in abundance (adjusted *p*-value <0.05 and a 2-fold change in abundance). Within the treatment, 50 proteins were increased in abundance, and 340 decreased (Appendix A).

Proteins found to be significantly decreased in abundance in the treatment group include Parkinson’s disease protein 7 (PARK7), elongation factor G, mitochondrial (GFM1), superoxide dismutase (SOD1), high mobility group protein HMG-I/HMG-Y, transcription factor A, mitochondrial (TFAM). The proteins significantly increased in abundance include dihydrofolate reductase (DHFR), sorting nexin-3 (SNX3) and s-formylglutathione hydrolase. Proteins related to neurodegenerative disease were also found to be significantly changed in abundance, including proteins involved in Parkinson’s disease (PARK7) and amyotrophic lateral sclerosis (ubiquilin-4, superoxide dismutase, and RNA-binding protein FUS).

Analysis of interactions between proteins changing in abundance was performed using STRING. Of the 50 proteins of increased abundance, 49 were able to be mapped within STRING. Statistical analysis identified an enrichment of pathways involved in the unfolded protein response and components of the proteostatic mechanisms of the cell. The STRING analysis of the proteins significantly increased in abundance can be found at the following permalink: https://version-11-0b.string-db.org/cgi/network?networkId=bXEIWsTDm0Bp (accessed on 4 March 2021). Of the 340 proteins decreased in abundance within treated cells, 316 were mapped within STRING; the analysis identified a decrease in proteins involved in mRNA metabolic processes and gene expression. Further KEGG pathway analysis of proteins decreasing in abundance showed proteins involved in thermogenesis and oxidative phosphorylation, and disease pathways including Alzheimer’s disease, Parkinson’s disease, and Huntington’s disease. Disease-annotated proteins are mainly mitochondrial proteins essential for mitochondrial homeostasis (NDUFA4, SOD1, TFAM, COX5A) or the prevention of apoptosis (BID and CYCS). The STRING analysis of the proteins significantly decreased in abundance can be found at the following permalink: https://version-11-0b.string-db.org/cgi/network?networkId=bjdJP4IT0l36 (accessed on 4 March 2021).

## 4. Discussion

In this study, we demonstrate the first use of the tyrosinase reaction to convert whole-sample peptide digests for subsequent proteomic analysis, providing spectral evidence for the incorporation of DOPA into the proteome of a neuronal human cell line. To the best of our knowledge, this is the first presentation of shotgun proteomic data analysed for DOPA as a protein constituent on a proteome scale and the first to provide spectral data generated by DDA to a public repository. These data also provide an example of where matching between runs (peptide identification transference) from a tyrosinase-converted proteome allowed identification of DOPA-containing sequences in biological samples (Table 2).

The initial trialing of tyrosinase as a conversion method utilised the method presented by Ito et al. [35], with successful conversion of a synthetic peptide from the protein UCP5 (SLC25A14) shown in Figure 2. This method was not successful in a whole proteome conversion (data not shown), so an alternative method was employed that in theory presented an efficiency of conversion of >95% on free tyrosine [34]. The Cieńska et al. (2016) [34] protocol needed to be scaled down to be economically feasible and accommodate the smaller sample material present in typical proteomic samples. The conversion of peptide residues to DOPA was achieved at a ~10% efficiency (Figure 3) regardless of the ratio of tyrosinase to peptide; these data indicate that factors other than the amount of tyrosinase are affecting the efficiency of conversion. The converted samples displayed an increase from 1% methionine oxidation within the control sample to ~75% across all three conversion samples indicating a high level of oxidation has occurred during conversion.

The conversion of proteins to contain L-DOPA is a pursuit within the field of synthetic biomaterials and has potential as a biological glue. Several pursuits of conversion of marine proteins from mussels have reported <15% yield on peptide conversions or 8% on specific residue conversion with tyrosinase [43]. Furthermore, the efficiency may be lower due to a loss of peptide solubility as it has been reported that use of strong acid conditions such as acetic acid can enhance the recovery of L-DOPA-containing proteins when separating from insoluble material [43]. Analysis of the amount of ionised peptide (Figure 4) agrees with the loss of peptides as amounts loaded were assumed on the amount of control peptide used for each conversion reaction. The decrease in ionisable peptides could be explained by several factors, including a loss of solubility, or some proteins possibly remained bound to the tyrosinase enzyme. It has been shown previously that the incorporation of L-DOPA into a non-solvent-exposed region of a protein can lead to protein insolubility, whilst external residues do not affect protein solubility [21]. Future improvements to the conversion experiment could include the use of a modified version of tyrosinase as the mtyr-CNK tyrosinase that is active at below 30 °C and optimally active at pH 7.0 may result in a marked increase in the specificity and reduction in oxidations found in this study [44].

There are several factors listed in the literature that could be leading to an increase in the amount of oxidation products within the sample. Tyrosinase isolates have been known to be contaminated with other oxidases [44], which could be leading to the off-target effects. L-DOPA may not have been the only product, there may have been 3,4,5-trihydroxyphenylalanine (TOPA) as has been reported in a 2002 study [45]. Any free tyrosine remaining in the lyophilised peptide sample may have been converted to L-DOPA, which can in turn increase the amount of ROS species. Similarly, the formation of L-DOPA in the peptide backbone could also be increasing the amount of ROS generated, especially if the tyrosinase powder contained impurities such as metals to allow Fenton chemistry to progress [25,46]. One of the then stated objectives of this work was the creation of a spectral library for use in DIA based experiments. The low conversion efficiency of tyrosine to DOPA prevents the use of these data in the construction of a comprehensive library and the optimisations listed above need to be performed before tyrosinase conversion becomes a viable method for the generation of spectral libraries for DIA experiments.

Treatment of the SH-SY5Y cells with 500 µM L-DOPA resulted in a significant increase in methionine oxidation (25%) (Figure 3) which is used in a proteome as a measure of oxidative stress [47]. However, L-DOPA treatment did not result in a statistically significant increase in the amount of DOPA- or oxidised phenylalanine-containing peptides which may be reflective of the neuronal cell line’s ability to turn over oxidised and damaged proteins [48] or the acute nature of the treatment.

Analysis of L-DOPA-containing sequences resulted in the identification of 37 (30 unique) proteins in the control and 75 (68 unique) in L-DOPA-treated cells. Analysis of the proteins containing L-DOPA unique to control via GO terms identified general terms including response to stress; organelle organization; post-transcriptional regulation of gene expression; and sulfur compound biosynthetic process. These do not highlight a specific group of proteins in the cell that would be ascribed to detrimental pathology but can be taken forward in literature as naturally occurring. Analysis of the proteins containing L-DOPA unique to treatment revealed four GO terms including protein folding; establishment of localisation; transport; and regulation of biological quality. These terms taken together show proteins that are responsible for the quality control of proteins are being either damaged by hydroxyl attack or are aggregating within the cell due to high levels of L-DOPA incorporation. From these experimental results, it is not possible to determine whether the L-DOPA proteins are indeed aggregating, which is a function of the protein half-lives, or whether it is a totally random pattern. Further experiments involving the use of the D-DOPA isomer may allow the distinction of ROS species generation and the effects of incorporation as this would still result in oxidative stress but not incorporation. The use of degradomics with stable isotope labels in pulse chase experiments during treatment may help identify the proteostatic influence due to L-DOPA treatment and the functions behind incorporation.

Pathway analysis within Mass Dynamics on the L-DOPA-treated samples showed a statistical significantly enrichment for RNA metabolism and processing pathways within proteins decreasing in abundance (see external project). A decrease in RNA expression is supported by the oxidative stress observed in Parkinson’s disease studies [49,50]. Within the treated sample, superoxide dismutase (SOD1) was decreased in abundance; this protein is involved in the scavenging of free radicals within the cell and acts to protect proteins from damage. A decrease in the amount of SOD1 would lead to a decreased capacity to cope with oxidative stress, and SOD1 has been found oxidatively damaged in Alzheimer’s disease and Parkinson’s disease patients [51].

Analysis of changes in protein abundance in STRING revealed that proteins involved in the unfolded protein response were increased due to treatment. Concurrently, proteins involved in transcription were down-regulated, suggesting that L-DOPA treatment results in cell stress related to protein misfolding. Several proteins involved in neurodegenerative diseases were significantly changed by L-DOPA treatment, namely those involved in mitochondrial homeostasis providing further evidence that L-DOPA may induce proteome changes that have previously been linked to neurodegenerative pathology [52,53,54,55]. As this study used an acute treatment, future proteomic analyses may benefit from chronic L-DOPA exposure. Exploring longer treatments may help to further understand the underlying mechanisms related to L-DOPA incorporation in proteins [7].

The experimental data demonstrate cell stress under L-DOPA treatment, which agrees with in vitro studies that have shown that L-DOPA can be toxic to dopaminergic neurons [16,56] and raises the possibility that L-DOPA could impair neuronal function in vivo. Progressive peripheral nerve deficits have been reported with increasing exposure to L-DOPA [57]; in a subset of L-DOPA-treated patients, a decrease in striatal DA transport binding was found, indicating a reduction in dopaminergic neuron function or number [58]. The toxic effects of L-DOPA in vitro can be divided into those due to oxidative stress, which is likely to be an in vitro phenomenon, and those that are due to an effect on protein, which is independent of oxidative stress [5,7]. These have been hypothesised to be a result of the mistaken incorporation of L-DOPA into proteins [17] which is further evidenced by the prevalence of misfolded protein responses identified within these experimental data.

This proteomic experiment was performed without fractionation or enrichment. It is foreseeable that highly fractionating the samples could increase the identification of L-DOPA-containing proteins. Furthermore, online/inline fractionations such as that of gas-phase fractionation can be performed on a pooled sample to enable matching between runs, which in essence is a manner for reproducing highly empirical DIA libraries [59]. Enrichment methodologies are routinely used in the PTM analysis field of phospho-proteomics [60,61] and the creation of either a chemical or antibody-based enrichment method for L-DOPA could then be applied to biological or tyrosinase-converted samples. The use of an antibody affinity method would require high specificity that would be expensive to raise; a chemical-based affinity method may offer more feasibility due to the unique catechol chemical structure of DOPA.

## 5. Conclusions

This work developed a microscale tyrosinase conversion method for the creation of DOPA-containing samples to aid in the exploration of its presence in the human proteome. This work also identified proteins containing DOPA generated in vitro via treatment with L-DOPA and an increase in the unfolded protein response with significantly changing proteins related to neurodegenerative diseases. Further optimisation experiments listed in the discussion need to be performed to create a sufficiently comprehensive spectral library for future DIA experiments.

## Figures and Tables

**Figure 1 proteomes-09-00024-f001:**
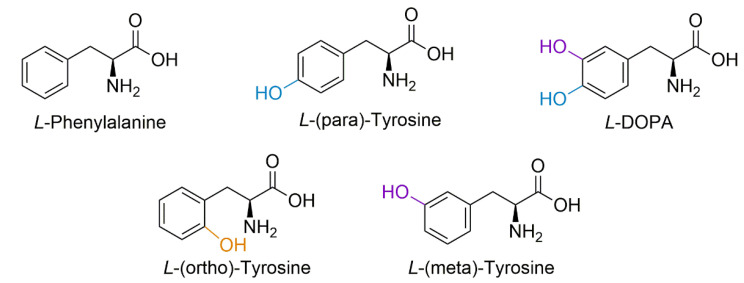
Structural forms of phenylalanine, para-tyrosine, meta-tyrosine, ortho-tyrosine, and L-DOPA. Hydroxyl radical attack on the phenyl ring of phenylalanine residues can form isomers of tyrosine, the canonical form being para-tyrosine, but can also form meta-tyrosine and ortho-tyrosine; hydroxyl attack of tyrosine can also form *L*-DOPA.

**Figure 2 proteomes-09-00024-f002:**
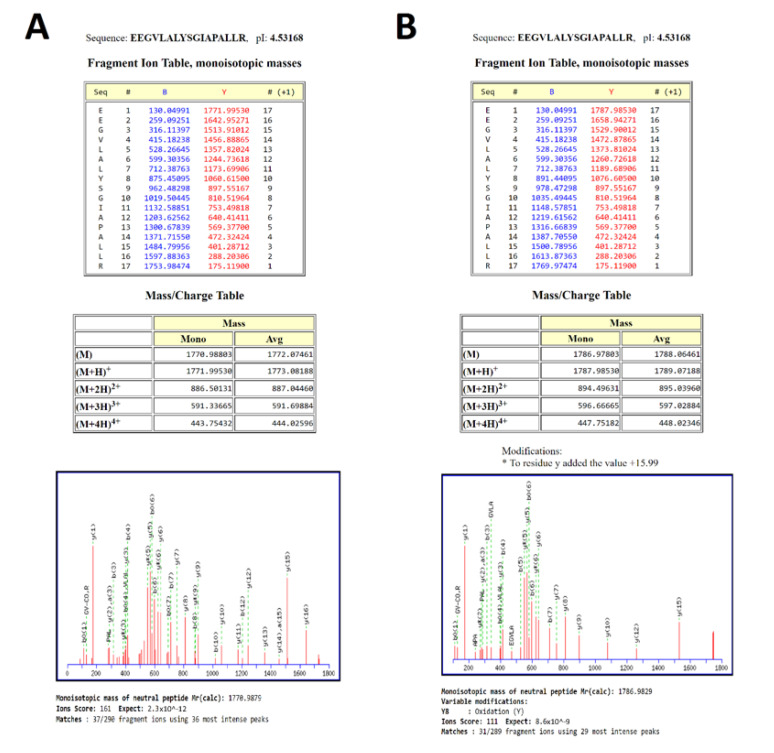
Spectral evidence of a positive control for PB-DOPA. (**A**) The native peptide sequence. (**B**) Addition of oxygen to tyrosine indicated by the increase in peptide mass and altered masses of the b8 and y10 ions.

**Figure 3 proteomes-09-00024-f003:**
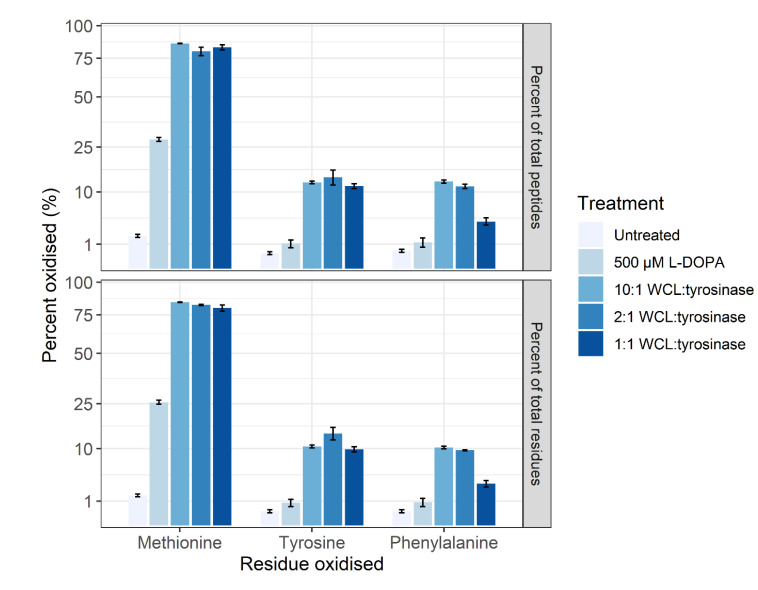
The overall oxidation profile of untreated, DOPA-treated and converted SH-SY5Y proteomes. The oxidation profile of methionine, tyrosine (L-DOPA), and phenylalanine is shown for each treatment group. The level of oxidation is presented as a percentage of the total number of identified peptides containing M, Y or F (**top**) and individual residues (**bottom**). The three biological control samples were pooled for the tyrosinase conversion, using tyrosinase to peptide digest ratios of 10:1, 2:1, and 1:1. Regardless of residue or peptide sequence, the percentage of oxidation follows a similar pattern for each sample. Error bars represent the standard error of the mean (SEM).

**Figure 4 proteomes-09-00024-f004:**
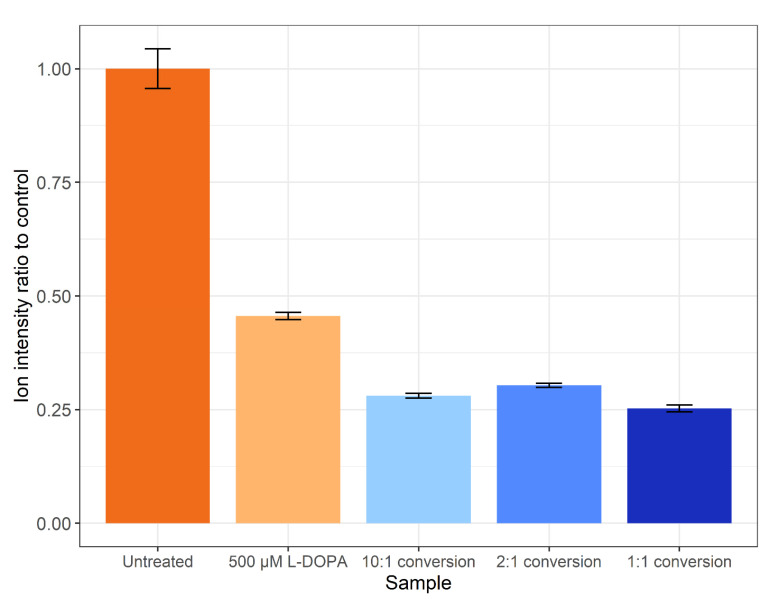
The ratio of total ion intensities compared to the control. Error bars represent the SEM.

**Figure 5 proteomes-09-00024-f005:**
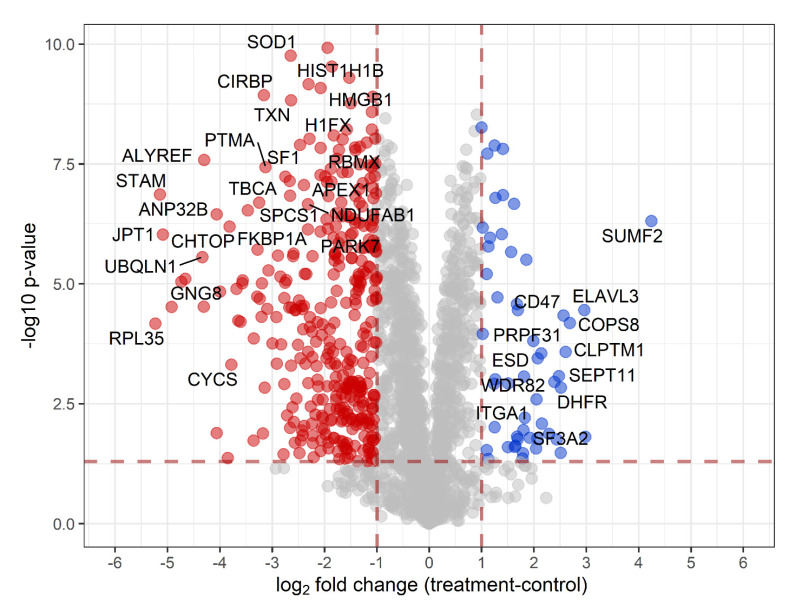
Volcano plot of proteins from label-free quantitative analysis. Coloured proteins have *p*-value <0.05 and a log_2_ fold change of ≥1. Blue proteins are up in treatment, and red proteins are down.

**Table 1 proteomes-09-00024-t001:** Peptide sequences containing DOPA within each and proteins identities matched (using accessions and UniProt ID matching).

Identification	Control	Treatment	Conversion
**peptides**	34	101	532
**proteins**	37	75	317

**Table 2 proteomes-09-00024-t002:** Peptide identifications due to conversion samples containing quality spectra. The four peptide sequences were only able to be identified due to the converted sample having a high-quality spectrum.

Localisation	Sequence	Protein Accession	PTM Site Ascore
**Control**	AAGGDGDDSLY(+15.99)PIAVLIDELR	P30154|2AAB_HUMAN	Y11:L-DOPA:1000.00
**DOPA**	DLYANTVLSGGTTMY(+15.99)PGIADR	P60709|ACTB_HUMAN:P63261|ACTG_HUMAN	Y15:L-DOPA:27.62
**DOPA**	GINPDEAVAY(+15.99) GAAVQAGVLSGDQ(+0.98)DTGDLVLLDVC(+71.04)PLTLGIETVGGVMTK	P11021|GRP78_HUMAN	Y10:L-DOPA:1000.00;Q23:Deamidation (NQ):0.00;C34:Propionamide:1000.00
**DOPA**	DLYAN(+.98)TVLSGGTTMY(+15.99) PGIADR	P60709|ACTB_HUMAN:P63261|ACTG_HUMAN	N5:Deamidation (NQ):1000.00;Y15:L-DOPA:10.83

## Data Availability

All project data are publicly available at PRIDE Identifier PXD025759, including FASTA file; PEAKS Studio Summary file; PEAKS Studio project files raw files; CSVs; and the MaxQuant search files. Pathway analysis project can be found at: https://app.massdynamics.com/p/dfbb55e9-0a6d-450e-8e8e-3f0c070e5bf9 (accessed on 1 March 2021).

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
