# Peer review of "A Novel Method for Creating a Synthetic L-DOPA Proteome and In Vitro Evidence of Incorporation"

_proteomes, 2021, doi:10.3390/proteomes9020024_

Round 1

Reviewer 1 Report

Steele et al., did excellent work in deciphering the new method for creating synthetic L-DOPA. I have the following queries to be addressed before final acceptance of the article

  1. Author modified the protein with L-DOPA. I was wondering that modified protein will retain its native structure? Did author perform any controlled experiment to look for its structural integrity?
  2. Did the author use a higher concentration of Urea to denature the protein or TCEP and acrylamide were sufficient? Please mention in the method section.
  3. Author should also include a material section to mention all the reagents/chemicals/peptide used in this study with catalog numbers

Author Response

Reviewer 1

Steele et al., did excellent work in deciphering the new method for creating synthetic L-DOPA. I have the following queries to be addressed before final acceptance of the article.

  • We appreciate your peer review and comments.
  1. Author modified the protein with L-DOPA. I was wondering that modified protein will retain its native structure? Did author perform any controlled experiment to look for its structural integrity?
  • Assessment of structural integrity changes due to L-DOPA incorporation has been performed previously by Ozawa et al 2015 which should have been mentioned in the discussion. Briefly, L-DOPA incorporation on surface/solvent-exposed sites does not result in a change of native conformation. If the incorporation is in a hydrophobic region inside the protein's structure, the protein will lose solubility.
  • The conversion did produce an insoluble pellet which was only able to be partially resolubilised. Preliminary analysis showed that peptides derived from this material contained DOPA-containing peptides. However, we have not included this data because we need to develop better resolubilisation techniques for this material. Once achieved, we can add this data to the spectral library and increase its comprehensiveness.
  • Some proteins from actual samples are going to be similarly insoluble and not analysable, requiring development of these solubilisation methods.
  1. Did the author use a higher concentration of Urea to denature the protein or TCEP and acrylamide were sufficient? Please mention in the method section.
  • The urea concentration has been updated within the manuscript; 8M Urea 2M Thiourea (supplemented with 1% C7bzO) was used for solubising the sample. It is the author's opinion that other non-traditional liquid buffer systems should be used for solubising L-DOPA containing proteins in future as a clear loss of protein has occurred. 5mM TCEP was sufficient for the breaking of the di-sulphide bridges but it remains undetermined if there are di-tyrosine bridges within the samples or a population of proteins unseen due to their insolubility.
  1. Author should also include a material section to mention all the reagents/chemicals/peptide used in this study with catalog numbers.
  • The materials section was populated with the relevant catalogue numbers.

  • Extensive edits have been made to the manuscript to improve readability, grammar and sentence structure. Substantial edits were made due to an error found in the production of figure 5. Section 3.4 was expanded to incorporate data analysis performed in STRING and a more in-depth analysis of proteins changing in abundance.(Lines 298-341, Lines 400-473)

Reviewer 2 Report

The manuscript “A novel method for creating a synthetic L-DOPA proteome and 2 in vitro evidence of incorporation.” by Steele JR et al. aims to establish the use of tyrosinase to convert proteomes, enabling subsequent analyses. The effects of DOPA on a neuronal cell line are investigated, as well as  the proteins that contain PB-DOPA by treatment with L-DOPA.

The idea is interesting, the manuscript is well written and this paper provides a method for enzymatically creating levodopa containing proteins using the enzyme tyrosinase with spectral evidence of in vitro incorporation. However, some relevant doubts still remain on the efficiency and appropriateness of technical solutions on the application of the method for the investigation of DOPA-treated and -untreated cells. More work is needed to make clear this out and the manuscript can be suggested for publication only if major revisions are satisfied. They are explained here below.

The authors discuss the use of using a data independent acquisition (DIA) method and the lack of sustainability (lines 54-60)  but do not compare neither discuss the expected performance nowadays using data dependent acquisition (DDA), which was used in this work. On that matter, it is fair to consider that the minimal depth of proteome analysis using DDA nowadays to be aimed to and reached – if one wants to assess a detailed quantitative and proteome-wide investigation using similar or even lower level instrumentation that those used in this work and few ours of LC-MS/MS analysis per sample- should be at least 5000 thousands proteins with at least two unique peptides per proteins, if not better (for example Pirmoradian M et al. Mol Cell Proteomics, 2013 or other many publications). Here the result come from the analysis of large culture dishes and cell number is not a problem, therefore the proteomics results of these analysis should be obtained at higher depth. The fact that only 2400 protein groups with 1 unique peptide have been found is an indication of not optimal conditions for LC-MS sample analysis and with potential loss of LC-MS performance and spectra quality.

The justification that peptide fractionation would bring higher depth is not sufficient as peptide fractionation would not allow for LFQ quantification over different samples, as that would be reliable only for unfractionated samples. This part is also not sufficiently treated in the discussion. The method should be optimized prior publication at least for the intentions of this manuscript.

-As major revision item, optimization of LC-MS analysis is to be performed and new results to be subjected to similar data analysis provided in the manuscript. The authors should provide results in quantitative proteomics filtered for at least 2 unique peptides per protein.

-In addition, the raw data should be provided by the authors to the journal to enable check of quality and raw file for review, which currently is not provided, PRIDE archive is only mentioned in “data availability” but the Id and details to retrieve data is missing. It is here required that PRIDE archive for proteomics data with complete raw data and archive credentials for reviewers, together with the description and complete parameter on how to reproduce final analysis from submitted raw data, will be provided to reach any positive evaluation of the manuscript.

-Related to similar arguments, it seems that statistical values have been arbitrarily picked for FDR less than 5% and a two-fold change in abundance is arbitrary. The authors should discuss why such values are to be considered the cut-off of relevance.  The figure 5 (volcano plot) is missing the label on the Y dimension in the plot. It is highly recommended that the authors use -Log10 transformation of the pValue.

-Principal Component Analysis (PCA) of results should be performed to determine if the control and treated samples cluster properly, if a general low level of reproducibility appears in results, possibly be related to unoptimized sample analysis workflow, a higher number of replicates is also suggested.

-Please also justify the following (line 523): “The inference was based on feature detection rather than area”

-Figure3: In the top panel, please specify the “X” label of the plot, what is represented by the histogram group on the left, center and right. Please also normalize all groups of histograms over the relative controls (with controls equal to 1 for each group of histograms related to M, Y and F oxidation).

-Figure4: Please normalize over the control (with control equal to 1)  

-Another relevant missing information in the manuscript, to be provided is the time of treatment of cells with DOPA, that is to be clearly specified in the manuscript. In how much time the proteins are regulated in abundance?

Other minor errors / typos in the text:

-Please eliminate double dot at the end of the figure 4 legend.

- Line 94: UTC7: Please define composition of the buffer and put the abbreviation in parentheses.

In conclusion the manuscript has an interesting idea, but that needs more work to find the optimal conditions in proteomics analysis and final quality of results or their representation. The manuscript is not recommended now for publication, which will be the case if the suggested revision items are successfully achieved.

Author Response

Reviewer 2

The manuscript "A novel method for creating a synthetic L-DOPA proteome and 2 in vitro evidence of incorporation." by Steele JR et al. aims to establish the use of tyrosinase to convert proteomes, enabling subsequent analyses. The effects of DOPA on a neuronal cell line are investigated, as well as the proteins that contain PB-DOPA by treatment with L-DOPA.

  • We appreciate your peer review and comments.

The idea is interesting, the manuscript is well written and this paper provides a method for enzymatically creating levodopa containing proteins using the enzyme tyrosinase with spectral evidence of in vitro incorporation. However, some relevant doubts still remain on the efficiency and appropriateness of technical solutions on the application of the method for the investigation of DOPA-treated and -untreated cells. More work is needed to make clear this out and the manuscript can be suggested for publication only if major revisions are satisfied. They are explained here below.

The authors discuss the use of using a data independent acquisition (DIA) method and the lack of sustainability (lines 54-60)  but do not compare neither discuss the expected performance nowadays using data dependent acquisition (DDA), which was used in this work. On that matter, it is fair to consider that the minimal depth of proteome analysis using DDA nowadays to be aimed to and reached – if one wants to assess a detailed quantitative and proteome-wide investigation using similar or even lower level instrumentation that those used in this work and few ours of LC-MS/MS analysis per sample- should be at least 5000 thousands proteins with at least two unique peptides per proteins, if not better (for example Pirmoradian M et al. Mol Cell Proteomics, 2013 or other many publications). Here the result come from the analysis of large culture dishes and cell number is not a problem, therefore the proteomics results of these analysis should be obtained at higher depth. The fact that only 2400 protein groups with 1 unique peptide have been found is an indication of not optimal conditions for LC-MS sample analysis and with potential loss of LC-MS performance and spectra quality.

  • We have edited the manuscript to reflect that a comprehensive spectral library was not successfully produced with methods (Lines 363-369).
    • "One of then stated objectives of this work was the creation of a spectral library for use in DIA based experiments. The low conversion efficiency of tyrosine to DOPA prevents the use of this data in the construction of a comprehensive library and the optimisations listed above need to be performed before tyrosinase conversion becomes a viable method for the generation of spectral libraries for DIA experiments."
  • The authors agree that the identified protein list does not reflect optimal conditions. However, this study aimed to assess the feasibility and provide a first case use of this methodology to the community in order for others to have a potential tool to study L-DOPA presence in proteomes. We did not at the outset intend on performing a comprehensive proteomic analysis. This data set was generated during the installation of our new Q-Exactive Plus platform, and methods were still under development. We do routinely get >5000 proteins quantified in experiments.

The justification that peptide fractionation would bring higher depth is not sufficient as peptide fractionation would not allow for LFQ quantification over different samples, as that would be reliable only for unfractionated samples. This part is also not sufficiently treated in the discussion. The method should be optimised prior publication at least for the intentions of this manuscript.

  • Fractionation can be performed in a reproducible fashion and methods have been reported. Ideally, most proteomic practitioners tend against this approach due to the increase in analysis time rather than a lack of reproducibility. Furthermore, online/inline fractionations such as that of Gas-Phase fractionation can be performed on a pooled sample to enable matching between runs, which in essence is a manner for reproducing highly empirical DIA libraries (Searle et al. 2020 Nature Comms). This has been updated in the text (Line 393)

-As major revision item, optimisation of LC-MS analysis is to be performed and new results to be subjected to similar data analysis provided in the manuscript. The authors should provide results in quantitative proteomics filtered for at least 2 unique peptides per protein.

  • Due to circumstances, the authors cannot perform the required experiment and optimisation within a ten-day turnaround. Furthermore, this experimental procedure aimed to provide proof of concept rather than a comprehensive proteome analysis. The authors have clarified that the MaxQuant search was performed using standard 2 minimum peptides for the LFQ component of the analysis.
  • The authors have performed a peptide centric analysis for modified species which is demonstrated within the Peaks Studio analysis workflow. If the aim was proteoform/protein centric analysis, 2D-PAGE/DIGE would have been employed. Arguments can be made that a peptide centric workflow will offer insight into potential proteoforms present but due to the nature of shotgun proteomics the actual proteoform is lost.

-In addition, the raw data should be provided by the authors to the journal to enable check of quality and raw file for review, which currently is not provided, PRIDE archive is only mentioned in "data availability" but the Id and details to retrieve data is missing. It is here required that PRIDE archive for proteomics data with complete raw data and archive credentials for reviewers, together with the description and complete parameter on how to reproduce final analysis from submitted raw data, will be provided to reach any positive evaluation of the manuscript.

  • These data have been prepared and were awaiting the feedback of the reviewers in case the inclusion of other prepared datasets would be required which would then require multiple submissions and usage of server bandwidth. Several issues with the PRIDE upload have occurred and until a PXD number is received we are providing a google drive link access for reviewers to use in the interim. https://drive.google.com/drive/folders/1xz7rRRFBJ4y7ktDuAR0g7JXgNmJiA_yH?usp=sharing

-Related to similar arguments, it seems that statistical values have been arbitrarily picked for FDR less than 5% and a two-fold change in abundance is arbitrary. The authors should discuss why such values are to be considered the cut-off of relevance.  The figure 5 (volcano plot) is missing the label on the Y dimension in the plot. It is highly recommended that the authors use -Log10 transformation of the pValue.

  • A new figure has been constructed according to this comment.
  • The data analysis has been performed in a more stringent manner for quantitation than a standard MaxQuant output with further analysis on the Mass Dynamic website using LIMMA. It was determined by the authors that in order to look at the broad dynamics of the effects on the proteome that using a wide FDR would indeed include false positives but would also provide more information for pathway analyses. The subsequent pathway investigation with the integrated Reactome-Volcano plot generation allowed the authors to manually investigate if proteins changing significantly had co-changing interaction partners. Further analysis of the significantly changing proteins (adjusted p.value <0.05) using STRING was performed providing more evidence for the changes found in the Mass Dyanamics analysis.

-Principal Component Analysis (PCA) of results should be performed to determine if the control and treated samples cluster properly, if a general low level of reproducibility appears in results, possibly be related to unoptimised sample analysis workflow, a higher number of replicates is also suggested.

  • This is available on the Mass Dynamics project page under QC metrics, which can be considered as supplementary data for this manuscript.

-Please also justify the following (line 523): "The inference was based on feature detection rather than area"

  • This has been justified in text. (line 270)
  • "The identification was based on feature detection and 'match between runs' or peptide identification transference rather than peptide identification in each injection, and sequences that could not be confidently attributed to a condition were discarded".

-Figure3: In the top panel, please specify the "X" label of the plot, what is represented by the histogram group on the left, center and right. Please also normalise all groups of histograms over the relative controls (with controls equal to 1 for each group of histograms related to M, Y and F oxidation).

-Figure4: Please normalise over the control (with control equal to 1) 

  • This has been updated.

-Another relevant missing information in the manuscript, to be provided is the time of treatment of cells with DOPA, that is to be clearly specified in the manuscript. In how much time the proteins are regulated in abundance?

  • This has been added in the materials and methods section. (line 103). "for 24 hours".

Other minor errors / typos in the text:

-Please eliminate double dot at the end of the figure 4 legend.

  • Deleted

- Line 94: UTC7: Please define composition of the buffer and put the abbreviation in parentheses.

  • Buffer has been defined in text. (Line 108)

In conclusion the manuscript has an interesting idea, but that needs more work to find the optimal conditions in proteomics analysis and final quality of results or their representation. The manuscript is not recommended now for publication, which will be the case if the suggested revision items are successfully achieved.

  • Once again we thank the reviewer for their time and effort in peer reviewing this manuscript. We agree that more optimal conditions can be found but due to the novelty of the idea and in order for it to reach the public domain we wish to proceed without the generation of a new dataset.
  • Extensive edits have been made to the manuscript to improve readability, grammar and sentence structure. Substantial edits were made due to an error found in the production of figure 5. Section 3.4 was expanded to incorporate data analysis performed in STRING and a more in-depth analysis of proteins changing in abundance.(Lines 298-341, Lines 400-473)

Reviewer 3 Report

The manuscript titled "A novel method for creating a synthetic L-DOPA proteome and in vitro evidence of incorporation. " by  Padula et al. describes a new method to determine the content of L-DOPA and related non-canonical amino assays in mammalian proteomes.

The new method is a modification of a previously  described approach (Cienska et al.) as stated by the authors. Of note, this modification allows the mass spec assessment of samples from commonly used cell model (here SH0SY5Y cells).

The manuscript is overall very clear and concise. The Materials and Methods section contains all the required details to follow each experimental steps and the ensuing data analysis.

The data is presented in a organized and structured manner. Also, the data seem properly interpreted.

The Introduction provides sufficient background information  to provide a strong rationale for this new method. A more extended discussion of the possible side-effects of L-DOPA treatment in PD might help.

Technical question:

The protein preparation includes a centrifugation step to eliminate insoluble protein conformers. It is quite plausible that oxidized and L-DOPA containing proteins form such insoluble conformers. Can the authors address this as a possible limitation or even add data to assess what proportion of proteins in the soluble fraction are modified/contain L-DOPA?

Author Response

Reviewer 3

The manuscript titled "A novel method for creating a synthetic L-DOPA proteome and in vitro evidence of incorporation. " by Steele et al. describes a new method to determine the content of L-DOPA and related non-canonical amino assays in mammalian proteomes.

The new method is a modification of a previously described approach (Cienska et al.) as stated by the authors. Of note, this modification allows the mass spec assessment of samples from commonly used cell model (here SH-SY5Y cells).

  • Thank you, it was indeed a methodology developed by Cienska et al. but that was only applied to free tyrosine and we think the novelty here lies in performing it on peptides.

The manuscript is overall very clear and concise. The Materials and Methods section contains all the required details to follow each experimental steps and the ensuing data analysis.

The data is presented in a organised and structured manner. Also, the data seem properly interpreted.

The Introduction provides sufficient background information to provide a strong rationale for this new method. A more extended discussion of the possible side-effects of L-DOPA treatment in PD might help.

  • Added a section on the side effects of L-DOPA treatment in PD. (Lines 445-456)

Technical question:

The protein preparation includes a centrifugation step to eliminate insoluble protein conformers. It is quite plausible that oxidised and L-DOPA containing proteins form such insoluble conformers. Can the authors address this as a possible limitation or even add data to assess what proportion of proteins in the soluble fraction are modified/contain L-DOPA?

  • The possible formation of insoluble protein conformations has previously been demonstrated by Ozawa et al 2015. It is the modification of internal non-solvent exposed tyrosine positions that results in the formation of insoluble proteoforms. The change of surface exposed positions is not detrimental to solubility.
  • We have noted the limitation within the discussion. (Lines 374-376)
    • "It has been shown previously that the incorporation of L-DOPA into a non-solvent exposed region of a protein can lead to protein insolubility, whilst external residues do not effect protein solubility."
  • We have also generated solubility fractionated data that shows the changes in protein solubility within the fractions. But these data need to be validated before publication, but briefly; Cells treated with L-DOPA were sequentially solubilised with increasing surfactant and chaotropic buffers. It was noted that proteins containing L-DOPA decreased in solubility noted by not only the sequence modification itself but the abundance of the protein/ORF tending to be higher in the low solubility fractions. It is also noted in these data that the total aggregating protein amount increases in L-DOPA treatments and it appears that the L-DOPA proteins have ether captured proteins via aggregation or the other proteins present have not been fully solubilised with the L-DOPA containing peptide sequences remaining insoluble. The proteins that have lost solubility in traditional buffer systems has led to the authors beginning work on new systems that can handle these highly insoluble proteins.

  • Extensive edits have been made to the manuscript to improve readability, grammar and sentence structure. Substantial edits were made due to an error found in the production of figure 5. Section 3.4 was expanded to incorporate data analysis performed in STRING and a more in-depth analysis of proteins changing in abundance.(Lines 298-341, Lines 400-473)

Round 2

Reviewer 2 Report

Unfortunately, the quality of proteomics is not considered sufficient to possibly suggest this manuscript for publication.

In addition, the authors have not considered seriously the indications provided in the first revision, many issues were left unsolved (see below).

Among them, the major problem of this manuscript is the data quality of the proteomics results, which leverages not a sufficient trust in  what is presented through the LC-MS-based proteomics outcomes.as reproducible data. The fact that the requested files for review are not available in official proteomics repository with sufficient information to retrieve them provided to the reviewers does not put a better light on this work.

The authors should really consider improving their proteomics workflows, as they have all suffcient instrumentation and biological models in their hands to reach good datasets, which is poorly missing. Minimal proteomics performance and data quality should always be addressed before to address any other data analysis and discussion on results.

The fact that the authors express in word their capability to reach the minimal requirement for a good proteomics on different experiments, without providing access to proteomics repository, attempting to reach  higher quality datasets, and even excluding another analysis by LC-MS is strong reason not to consider the manuscript for publication (in this or any  proteomics journal). The indication of a potential critical problem in sample processing is also there, however this proteomics analysis would not be able to say more on this and the authors do not appear willing to discover more on this aspect.  

In the light of the above this manuscript is not recommended for publication.

Point-by-point replies to the authors are given below:

Authors´ answer to first revision:

 We have edited the manuscript to reflect that a comprehensive spectral library was not successfully produced with methods (Lines 363-369).

    • "One of then stated objectives of this work was the creation of a spectral library for use in DIA based experiments. The low conversion efficiency of tyrosine to DOPA prevents the use of this data in the construction of a comprehensive library and the optimisations listed above need to be performed before tyrosinase conversion becomes a viable method for the generation of spectral libraries for DIA experiments."

Reviewer 2. Sorry, but this insertion is not present in the new version of the manuscript (v2). The issue still unresolved.

_________________________

Authors´ answer to first revision:

  • The authors agree that the identified protein list does not reflect optimal conditions. However, this study aimed to assess the feasibility and provide a first case use of this methodology to the community in order for others to have a potential tool to study L-DOPA presence in proteomes. We did not at the outset intend on performing a comprehensive proteomic analysis. This data set was generated during the installation of our new Q-Exactive Plus platform, and methods were still under development. We do routinely get >5000 proteins quantified in experiments.

Reviewer 2. I understand, but the authors claim to be able to study and to interpret also the effects of DOPA on a neuronal cell line as they show data analysis of such proteomics data. The claim is clear when plots are made and interpretation of result is suggested, but on a poor dataset, which are clearly missing on sufficient proteome depth of analysis, given the material and instrumentation used. The data is still not available to reviewers as the PRIDE identifier is not provided in the V2 of the manuscript. It is still a question opened if the low depth is due to the poor quality of spectra or some problems in the sample handling. The authors admit that the conditions and parameters of analysis were not good, without specifying what got compromised. The study should fulfill minimal requirements to draw conclusions from proteomics results based on current technology. As the authors are able to get “routinely  >5000 proteins quantified in experiments” it is even more required that they re-analyze data with proper parameters and conditions to obtain better data quality and higher depth, analyzing and drawing conclusions on data with after having at least >5000 proteins quantified. This is quite a common and qualitative standard today in the field. This would be required as major revision for publication, which is not recommended at this state.

­­­­­­­­­­­­­­­­­­­­__________________________

Authors´ answer to first revision:

  • Fractionation can be performed in a reproducible fashion and methods have been reported. Ideally, most proteomic practitioners tend against this approach due to the increase in analysis time rather than a lack of reproducibility.

Reviewer 2. The authors should reference which method they refer. However one method was recently published on this matter but very few or no studies at all were  successful using such method. In the opposite case, the authors should reference studies from other groups using fractionation and label free quantification together. If not possible, as no studies have made successful use of it, a different approach than label-free quantification should be considered for the intentions of the study approached in this manuscript.  

Authors´ answer to first revision:

  • Furthermore, online/inline fractionations such as that of Gas-Phase fractionation can be performed on a pooled sample to enable matching between runs, which in essence is a manner for reproducing highly empirical DIA libraries (Searle et al. 2020 Nature Comms). This has been updated in the text (Line 393)

Reviewer 2. Again, the authors refer to an insertion that does not exist at line 393 of the manuscript V2 provided. Have the authors uploaded the right file? This issue stays unresolved.

__________________________

Authors´ answer to first revision:

  • Due to circumstances, the authors cannot perform the required experiment and optimisation within a ten-day turnaround. Furthermore, this experimental procedure aimed to provide proof of concept rather than a comprehensive proteome analysis. The authors have clarified that the MaxQuant search was performed using standard 2 minimum peptides for the LFQ component of the analysis..

Reviewer 2. Unfortunately this is not a sufficient answer to avoid minimal proteomics data quality. The proteomics analysis if used for publication for drawing conclusions  needs to fulfill minimal performance and data quality, relatively to the field of application. The request of perform new LC-MS analysis to provide minimal quality of results stays. As this cannot be done accordingly to the authors, who claim to be capable of it routinely, then the manuscript is not recommended for publication without sufficiently good results.

As the MaxQuant search is done in 2 minimum peptides for the LFQ component of the analysis, the authors should use a cut off of at least 2 unique peptide proteins for confidently identified proteins, while the others should be discarded.

__________________________

Authors´ answer to first revision:

  • These data have been prepared and were awaiting the feedback of the reviewers in case the inclusion of other prepared datasets would be required which would then require multiple submissions and usage of server bandwidth. Several issues with the PRIDE upload have occurred and until a PXD number is received we are providing a google drive link access for reviewers to use in the interim. https://drive.google.com/drive/folders/1xz7rRRFBJ4y7ktDuAR0g7JXgNmJiA_yH?usp=sharing

Reviewer 2. The PRIDE identifier, which is a number starting with “PDX” is never provided in the manuscript. Again, the authors should check the file provided for second revision. Google drive is not a proper proteomics repository. This issue stays unresolved.

__________________________

Authors´ answer to first revision:

  • This has been justified in text. (line 270)
  • "The identification was based on feature detection and 'match between runs' or peptide identification transference rather than peptide identification in each injection, and sequences that could not be confidently attributed to a condition were discarded".

Reviewer 2. Noted

__________________________

Authors´ answer to first revision:

  • This has been updated.

Reviewer 2. This answer is referred to figure 3 and figure 4 where the data should have been normalized on controls. This was not achieved in reality, and the figures are not including what asked. Controls would be 1 otherwise. This issue stays unsolved.

__________________________

Authors´ answer to first revision:

  • This has been added in the materials and methods section. (line 103). "for 24 hours".

Reviewer 2. Noted

Author Response

Reviewer 2

Unfortunately, the quality of proteomics is not considered sufficient to possibly suggest this manuscript for publication.

In addition, the authors have not considered seriously the indications provided in the first revision, many issues were left unsolved (see below).

Among them, the major problem of this manuscript is the data quality of the proteomics results, which leverages not a sufficient trust in  what is presented through the LC-MS-based proteomics outcomes.as reproducible data. The fact that the requested files for review are not available in official proteomics repository with sufficient information to retrieve them provided to the reviewers does not put a better light on this work.

The authors should really consider improving their proteomics workflows, as they have all suffcient instrumentation and biological models in their hands to reach good datasets, which is poorly missing. Minimal proteomics performance and data quality should always be addressed before to address any other data analysis and discussion on results.

The fact that the authors express in word their capability to reach the minimal requirement for a good proteomics on different experiments, without providing access to proteomics repository, attempting to reach  higher quality datasets, and even excluding another analysis by LC-MS is strong reason not to consider the manuscript for publication (in this or any  proteomics journal). The indication of a potential critical problem in sample processing is also there, however this proteomics analysis would not be able to say more on this and the authors do not appear willing to discover more on this aspect.  

In the light of the above this manuscript is not recommended for publication.

Point-by-point replies to the authors are given below:

Authors´ answer to first revision:

 We have edited the manuscript to reflect that a comprehensive spectral library was not successfully produced with methods (Lines 363-369).

  • "One of then stated objectives of this work was the creation of a spectral library for use in DIA based experiments. The low conversion efficiency of tyrosine to DOPA prevents the use of this data in the construction of a comprehensive library and the optimisations listed above need to be performed before tyrosinase conversion becomes a viable method for the generation of spectral libraries for DIA experiments."

Reviewer 2. Sorry, but this insertion is not present in the new version of the manuscript (v2). The issue still unresolved.

Author response: This insertion was and is present in the new version of the manuscript, however, there appears to have been a discrepancy in the line numbering between versions. It is present at lines 398-402.

_________________________

Authors´ answer to first revision:

  • The authors agree that the identified protein list does not reflect optimal conditions. However, this study aimed to assess the feasibility and provide a first case use of this methodology to the community in order for others to have a potential tool to study L-DOPA presence in proteomes. We did not at the outset intend on performing a comprehensive proteomic analysis. This data set was generated during the installation of our new Q-Exactive Plus platform, and methods were still under development. We do routinely get >5000 proteins quantified in experiments.

Reviewer 2. I understand, but the authors claim to be able to study and to interpret also the effects of DOPA on a neuronal cell line as they show data analysis of such proteomics data. The claim is clear when plots are made and interpretation of result is suggested, but on a poor dataset, which are clearly missing on sufficient proteome depth of analysis, given the material and instrumentation used. The data is still not available to reviewers as the PRIDE identifier is not provided in the V2 of the manuscript. It is still a question opened if the low depth is due to the poor quality of spectra or some problems in the sample handling. The authors admit that the conditions and parameters of analysis were not good, without specifying what got compromised. The study should fulfill minimal requirements to draw conclusions from proteomics results based on current technology. As the authors are able to get “routinely  >5000 proteins quantified in experiments” it is even more required that they re-analyze data with proper parameters and conditions to obtain better data quality and higher depth, analyzing and drawing conclusions on data with after having at least >5000 proteins quantified. This is quite a common and qualitative standard today in the field. This would be required as major revision for publication, which is not recommended at this state.

Author response: We initially chose to present the most robust data. In the manuscript we had stated the number of top proteins (as supported by the most number of unique peptides) identified in each protein group, which was a total of 4058 top proteins. However, the total number of identifications in the oxidative analysis was actually 9462 proteins, which is a more than sufficient proteomic depth; we have updated this in the manuscript to make the quality and depth of our data clear (lines 206-207).

The purpose of this manuscript is to provide a proof-of-concept for the microscale conversion of a proteome. A label-free quantitative analysis is provided as a supplement to correlate significant proteomic changes with the increase in protein oxidation demonstrated. For the quantitative analysis, MaxQuant identified 2047 protein groups (~8721 individual proteins) with ≥2 unique peptides, and so the depth of the two analyses are both sufficient. We also quantified 1846 total protein groups, which we have also added to the manuscript (line 220). Hence, the depth of our data is more than sufficient for publication.

As an aside, while we agree that the identification of >5000 proteins is certainly common using today’s methods, and while we were also able to identify >5000 proteins, it is not a hard rule. We note that a recent quantitative proteomic analysis published where <5000 proteins were identified using SH-SY5Y cells, the cell line utilised in our manuscript: Kern et al. 2021, Cells 10(2):404 identified 2334, 2933, and 4421 total proteins in three separate experiments.

­­­­­­­­­­­­­­­­­­­­__________________________

Authors´ answer to first revision:

  • Fractionation can be performed in a reproducible fashion and methods have been reported. Ideally, most proteomic practitioners tend against this approach due to the increase in analysis time rather than a lack of reproducibility.

Reviewer 2. The authors should reference which method they refer. However one method was recently published on this matter but very few or no studies at all were  successful using such method. In the opposite case, the authors should reference studies from other groups using fractionation and label free quantification together. If not possible, as no studies have made successful use of it, a different approach than label-free quantification should be considered for the intentions of the study approached in this manuscript.  

Author response: In response to the reviewer the use of labelling strategies drowns signal from unique PTMs. We have found this especially true through meta-analyses of Parkinson’s TMT-labelled datasets. As such, labelling strategies without an appropriate enrichment method (i.e., phosphorylation or TiO2) are fruitless. Also, the utilisation of FAIMS/IMS fractionation has been successfully used to increase LFQ quantitative accuracy and ID depth: Hebert et al. 2018, Analytical Chemistry 90(15):9529-9537. Use of fractionated DDA samples for matching between runs has been employed inline with Boxcar, this can of course be deemed a different acquisition strategy but in our eyes, we see this as a form of gas phase fractionation that can increase the number of ions identified within runs that can subsequently be used for LFQ (Meier et al. 2018, Nature Methods, 15(6):440-448). Furthermore, the vast majority of papers employing DIA based LFQ utilise fractionation techniques, one such technique is from the Searle Lab which does indeed use gas phase fractionation to enhance ID and quantitative depth: Searle et al. 2020, Nature Communications, 11:1548.

__________________________

Authors´ answer to first revision:

  • Furthermore, online/inline fractionations such as that of Gas-Phase fractionation can be performed on a pooled sample to enable matching between runs, which in essence is a manner for reproducing highly empirical DIA libraries (Searle et al. 2020 Nature Comms). This has been updated in the text (Line 393)’

Reviewer 2. Again, the authors refer to an insertion that does not exist at line 393 of the manuscript V2 provided. Have the authors uploaded the right file? This issue stays unresolved.

Author response: Again, this insertion was and is present in the new version of the manuscript, but there seems to have been a discrepancy in the line numbering between versions. The insertion is present at lines 460-462.

__________________________

Authors´ answer to first revision:

  • Due to circumstances, the authors cannot perform the required experiment and optimisation within a ten-day turnaround. Furthermore, this experimental procedure aimed to provide proof of concept rather than a comprehensive proteome analysis. The authors have clarified that the MaxQuant search was performed using standard 2 minimum peptides for the LFQ component of the analysis..

Reviewer 2. Unfortunately this is not a sufficient answer to avoid minimal proteomics data quality. The proteomics analysis if used for publication for drawing conclusions  needs to fulfill minimal performance and data quality, relatively to the field of application. The request of perform new LC-MS analysis to provide minimal quality of results stays. As this cannot be done accordingly to the authors, who claim to be capable of it routinely, then the manuscript is not recommended for publication without sufficiently good results.

As the MaxQuant search is done in 2 minimum peptides for the LFQ component of the analysis, the authors should use a cut off of at least 2 unique peptide proteins for confidently identified proteins, while the others should be discarded.

Author response: We reiterate that we identified 9462 proteins, which is more than sufficient. Revisiting the analysis, 7198 proteins were identified with 2 unique peptides, which is also more than sufficient. However, we disagree that use of 2 minimum peptides is a requirement for protein identification. One unique peptide was used for the PTM analysis as the purpose of this experiment was the identification of peptides, not proteins. Discarding positive oxidation IDs simply because one unique peptide was identified for a protein is counter-productive and arbitrary for the purpose of this analysis.

We also note the results of Gupta & Pevzner 2009, Journal of Proteome Research 8(9):4173-4181, which demonstrated that the “two-peptide” rule increases false discovery rates and reduces protein IDs to below the decoy database; it was indicated that protein ID should be based on error rates, not by number of unique peptides. Many proteomic analysis software use default settings of one unique peptides for protein identification, including PEAKS and MaxQuant respectively, which were both used in this manuscript.

We agree that for quantitation, a more confident analyses using a minimum of 2 unique proteins should be performed. We have re-performed our analysis with ≥2 unique peptides specified in MaxQuant for protein identification, prior to the Mass Dynamics analysis, to ensure no flow through of less confident proteins. We have clarified our settings for the analysis in the manuscript (lines 213-214) and updated the quantitative data. Only confidently identified proteins with a minimum 2 unique peptides are presented in the quantitative results.

__________________________

Authors´ answer to first revision:

  • These data have been prepared and were awaiting the feedback of the reviewers in case the inclusion of other prepared datasets would be required which would then require multiple submissions and usage of server bandwidth. Several issues with the PRIDE upload have occurred and until a PXD number is received we are providing a google drive link access for reviewers to use in the interim. https://drive.google.com/drive/folders/1xz7rRRFBJ4y7ktDuAR0g7JXgNmJiA_yH?usp=sharing

Reviewer 2. The PRIDE identifier, which is a number starting with “PDX” is never provided in the manuscript. Again, the authors should check the file provided for second revision. Google drive is not a proper proteomics repository. This issue stays unresolved.

Author response: The reviewer appears to have misinterpreted the purpose of the Google Drive link. A PRIDE identifier has been provided since the revised manuscript was initially submitted, and the authors were informed that this had been passed onto the reviewers. As previously stated, when the revised manuscript was submitted, the data had been uploaded to PRIDE however an identifier was not generated prior to the revision deadline. The google drive link was a temporary solution to provide the reviewers with interim access to the data. We will re-iterate here that the data can be accessed using the PRIDE identifier PXD025529. It has also been inserted in the manuscript twice (lines 210 and 486).

__________________________

Authors´ answer to first revision:

  • This has been justified in text. (line 270)
  • "The identification was based on feature detection and 'match between runs' or peptide identification transference rather than peptide identification in each injection, and sequences that could not be confidently attributed to a condition were discarded".

Reviewer 2. Noted

__________________________

Authors´ answer to first revision:

  • This has been updated.

Reviewer 2. This answer is referred to figure 3 and figure 4 where the data should have been normalized on controls. This was not achieved in reality, and the figures are not including what asked. Controls would be 1 otherwise. This issue stays unsolved.

Author response: The reviewer appears to be misinterpreting the figures.

Figure 3: This is a grouped bar graph and not a histogram. This should be clear as the data is presented as a percentage, not a frequency. Therefore, it would be incorrect to normalise the data to the control as this figure (as stated in the manuscript) demonstrates the overall oxidation level of each sample, not the level of oxidation relative to the control. The control whole cell lysate has a baseline oxidation level, and each treatment or conversion affects the overall percentage of oxidation differently; it is important to show these percentages rather than normalise the data to arbitrary values. We have updated the in-figure formatting and labelling and edited the figure legend to allow for easier interpretation for the reader.

Figure 4: The data was normalised to the control, as stated in the figure legend. The control has been re-inserted into the Figure to make it clear to the reader that the control does indeed equal ‘1’.

__________________________

Authors´ answer to first revision:

  • This has been added in the materials and methods section. (line 103). "for 24 hours".

Reviewer 2. Noted